# Sexual assault and abuse committed against family members: An analysis of 1342 legal outcomes and their motivations

Alberto Blandino[1]*, Lidia Maggioni[1], Francesca Chiaffarino[2], Fabio Parazzini[3], Daniele Capano[4], Elena Maria Florio[1], Manuela Margherita[1], Gian Marco Bertelle[1], Lorenzo Franceschetti[1], Alberto Amadasi[1], Giulia Vignali[1], Barbara Ciprandi[1], Graziano Domenico Luigi Crudele[1], Vera Gloria Merelli[1], Federica Collini[1], Enrico Angelo Muccino[1], Patrizio Nicolò[5], Giussy Barbara[6], Alessandra Kustermann[6], Cristina Cattaneo[1], Andrea Gentilomo[1]

1 Department of Biomedical Science for Health, Institute of Legal Medicine, Università Degli Studi, Milan, Italy, 2 Department of Woman, Newborn and Child, Gynaecology Unit, Fondazione Istituto di Ricovero e Cura a Carattere Scientifico (IRCCS) Ca' Granda Ospedale Maggiore Policlinico, Milan, Italy, 3 Department of Clinical Sciences and Community Health, Università Degli Studi, Milan, Italy, 4 U.O.C. Medicina Legale–ASST Lecco, Lecco, Italy, 5 Lawyers' Order of Milan, Milano, Italy, 6 Department of Obstetrics and Gynecology and Service for Sexual and Domestic Violence (SVSeD), Fondazione IRCCS Ca' Granda, Ospedale Maggiore Policlinico, Milan, Italy

* alberto.blandino@gmail.com

**Data Availability Statement:** All relevant data are within the paper and its Supporting Information files.

## Abstract

### Background

Over the past years medical centres specifically addressed in gender-based violence have developed protocols for the collections of evidence useful in the courtroom, including accurate documentation of physical and psychological states of the victim and collection of samples. Previous studies showed an association between documented physical trauma and conviction but unfortunately, few studies in the recent literature analysed the factors that influence the legal outcome and final judgement. The present study focused on the elements that appeared of significance in the legal outcome, including medico-legal evaluation, source of the crime report and circumstance of the assault.

### Methods

It was conducted a retrospective analysis of all the judgments issued by the Public Prosecutor's Office at a Court of a Metropolitan Italian city regarding sexual and domestic violence, from January 1st 2011 to 31st December 31st 2015. Examination regarded the demographic information of the victim and of the defendant, information on the crime, the circumstances of the aggression and medical information retrieved. Sentences were subsequently divided into two categories based on the legal outcome (conviction vs acquittal) and the different characteristics of the two sub-populations were compared to verify if there were variables significantly associated to the judge's final judgment.

**Funding:** The authors received no specific funding for this work.

**Competing interests:** The authors have declared that no competing interests exist.

## Results

Over the 5 years taken into consideration, there have been 1342 verdicts regarding crimes of sexual violence (374 cases) and regarding abuses against family members or cohabitants (875): other 93 cases regarded both sexual violence and abuse. 66.3% ended in conviction of the offender and 33.7% in acquittal of the accused. Cases of conviction were more frequent when they involved: use of a weapon by the assailant, as well as if the assailant had a criminal record and had a history of drug abuse or other addictions; duration of proceeding less 22 months and a civil party involved; presence of clinical documentation together with other deposition in addition to victim's deposition; also frequent episodes of violence and application of precautionary measures were associated to conviction.

## Conclusions

Many factors seem able to influence the judge's judgment, although clearly each case must be singularly evaluated. The mere presence of medical documentation, without the support of other sources of evidence, such as the victim's statement or further declarations, however, is almost always not definitive for the verdict. Despite so, in cases where there are multiple sources of evidence, clinical documentation can provide useful elements and can give clues on the consistency between the history told and injuries observed.

## Introduction

The Declaration on the Elimination of Violence against Women, adopted by the United Nations (1993), defines violence against women as "any act of gender-based violence that results in, or is likely to result in, physical, sexual or psychological harm or suffering to women, including threats of such acts, coercion or arbitrary deprivation of liberty, whether occurring in public or in private life" [1].

In Europe, the Council of Europe Convention on preventing and combating violence against women and domestic violence, the so-called Istanbul Convention (2011), confirmed the previous definition and defines gender-based violence as any violence directed against a woman as such, or that affects women disproportionately (art.3) [2].

By applying these definitions, different forms of violence against women can be identified, such as violence inflicted by partners, sexual violence imposed by non-partners, sexual harassment and violence in the workplace, educational institutions and in sport, genital mutilation, early or forced marriage, physical, psychological and economic abuse, etc.

According to Italian law, violence against women can be punished mainly on the basis of two articles of the Penal Code: one in which "anyone, by constraint, violence, deceit or abuse of authority, forces another person to commit or undergo sexual acts" (Penal Code, Art 609bis), thus including rape and sexual harassment, and one in which a person "abuses or maltreats a person of the family or otherwise cohabiting, or a person subject to his authority" (Penal Code, Art 572), including neglect and all cases of physical, psychological and economical violence repeated over time.

In Italy, according to the data of the latest survey published by ISTAT (Istituto Nazionale di Statistica—National Statistical Institute) (2014) [3], although the number of women who report having experienced violence (physical, sexual or also psychological) in the last 5 years has decreased compared to the previous ISTAT survey produced in 2006 [4], domestic and

sexual violence is still a widespread phenomenon, which has manifestations perceived as very serious by more women than in the past (34.5% of them feared for their lives, compared to 18.8% in 2006) and which are also, objectively, of greater physical severity, since in 40.2% of cases the victim suffered injuries, compared to 26.3% in 2006.

In November 2017, a National Anti-Violence Plan (2017–2020) was approved by the Italian Council of Ministers. This document was the result of an ad hoc working group attended by ministries, law enforcement agencies, regions and municipalities, trade unions and associations working on the theme of Violence [5]. Following the principles laid down in the Istanbul Convention, the Plan provides three major lines of action:

1. Preventing violence (through educational and communication plans, as well as training of public and private sector workers);

2. Protecting and supporting victims (through anti-violence territorial networks);

3. Prosecuting and punishing (in synergy with institutional agencies such as, for example, the Ministry of the Interiors and the Ministry of Justice).

Over the past years, worldwide, medical centres specifically addressed in gender-based violence have developed protocols for the collections of evidence useful in the courtroom, including accurate documentation of physical and psychological states of the victim, in addition to specific collection of samples. Previous studies showed an association between documented physical trauma and conviction [6–9] but unfortunately, few studies in the recent literature analysed the factors that influence the legal outcome and final judgement [10–15].

For this reason, the present study aims to describe the characteristics of judgements of the Court of First Instance of a Metropolitan Italian city, in particular by focusing on the elements of significance in the legal outcome, including medico-legal evaluation, source of the crime report and circumstance of the assault. Both crimes concerning sexual and domestic violence were analysed, considering the possible contribution that a detailed medical examination can give in such cases. A preliminary specific analysis on victims examined in a specialized centre was made, in order to improve medico-legal care quality; as previously stated, indeed, physicians who take part in these cases often have no feedback on the legal consequences of their intervention [16] and in recent years few studies worldwide have addressed medico-legal findings and epidemiological data impact on legal outcome [16–20].

## Materials and methods

The study was conducted through a retrospective analysis of all the judgments issued by the Public Prosecutor's Office at the Court of Milan over a 5-year period, from January 1st 2011 to 31st December 31st 2015. Data was provided anonymized by the court and is not publicly available. Only verdicts issued by the Court of First Instance, regarding sexual and domestic violence were selected, dividing those concerning crimes of sexual violence (Art. 609bis, Penal Code) from those concerning abuse committed against family members, cohabitants or person subjected to the authority of the alleged perpetrator (Art. 572 Penal Code). The judgements obtained were examined with a special focus on the demographic information of the victim (sex, age, nationality, occupation, use of drugs and/or alcohol) and of the defendant (nationality, type of relationship with the victim, use of drugs and/or alcohol, criminal record); information on the crime (type of offense, source of the crime report, type of evidence, duration of the proceeding, type of judgment, period between fact and issue of the sentence, sentence punishment, application of precautionary and personal security measures); the circumstances of the aggression (number of suspects, presence of minors, place of the offense, vaginal/oral/anal

penetration in case of sexual violence, use of weapons, use of objects, frequency of episodes) and medical information (access of the victim in a emergency room or in a specialized centre, time elapsed between the violence and the access, prognosis (in days) assigned in Hospital, the presence of physical injuries–type [abrasions, bruises, lacerations, fractures, burns and combination of the above] and location [head, neck, thorax, abdomen, upper limbs, lower limbs, multiple locations]–and finally need for surgical interventions) (Table 1).

The Milan Court encompasses a large area, not only from the provincial territory, but also victims from nearby provinces. In such a territory the city has a centre open 24 hours a day [called *Soccorso Violenza Sessuale e Domestica* (SVSeD)], specialized in the comprehensive care (including gynaecology-obstetrics, psychology and medico-legal staff) and support of sexual and domestic violence victims of all kinds; this Centre is established in the Fondazione IRCCS Ca'Granda Ospedale Maggiore Policlinico of Milan. Each case that reaches the attention of the SVSeD is routinely investigated with complete medical history and clinical examination, performed with the consent of the victim, using colposcopy in sexual assault victims; biological trace evidence is also collected when indicated. A focus on cases both present in the SVSeD records and in archives of the First instance Court was made.

**Table 1. Information extrapolated from judgments.**

| | |
|---|---|
| **Demographic information of the victim** | sex |
| | age |
| | nationality |
| | occupation |
| | use of drugs and/or alcohol |
| **Demographic information of the defendant** | nationality |
| | type of relationship with the victim |
| | use of drugs and/or alcohol |
| | criminal record |
| **Information on the crime** | type of offense |
| | source of the crime report |
| | type of evidence |
| | duration of the proceeding |
| | type of judgment |
| | period between fact and issue of the sentence |
| | sentence punishment |
| | application of precautionary |
| | Application of personal security measures |
| **Circumstances of the aggression** | number of suspects |
| | presence of minors |
| | place of the offense |
| | vaginal/oral/anal penetration in case of sexual violence |
| | use of weapons |
| | use of objects |
| | frequency of episodes |
| **Medical information** | access of the victim in emergency room or in a specialized centre |
| | time elapsed between the violence and the access |
| | prognosis days assigned in Hospital |
| | the evidence of injuries (typology, location) |
| | necessity of surgical interventions |

Sentences were subsequently divided into two categories based on the legal outcome (conviction/acquittal) and the different characteristics of the two sub-populations were compared to verify if there were variables significantly associated to the judge's final judgment.

## Statistical analysis

The association between selected variables and a conviction were computed using unconditional logistic regression models and expressed as odd ratio (OR) and their 95% confidence interval (CI). We computed an univariate unconditional logistic regression model for each variable and reported unadjusted ORs and 95% confidence intervals. To account for potential confounding factors, we computed multivariate unconditional logistic regression models. We applied, to the overall sample, the backward step-wise selection method starting from a saturated model that included all the variables involved. The final model included the subsequent variables: defendant's addiction, criminal record, application of precautionary measures, type of evidence, duration of the proceeding, civil party, use of weapons, frequency of episodes.

We also computed multivariate logistic regression analyses in the two subgroups of domestic violence and sexual violence. To allow the comparison of the results we included in the multivariate logistic models, of the two subgroups, the same variables of the final model of the overall sample.

Result were considered significant at p-value < 0.05. All statistical analyses were carried out using the SAS statistical software package, version 9.4 (SAS Institute, Inc., Cary, NC, USA).

# Results

Over the 5 years taken into consideration, there have been 14334 verdicts regarding all crimes without any distinction: those concerning crimes of sexual violence (Art. 609bis, Penal Code) were 374 (27.9%), abuses against family members or cohabitants (Art. 572 Penal Code) were 875 (65.2%) and 93 (6.9%) cases where there was both sexual violence and abuse for a total of 1342 verdicts.

Globally, 66.3% (n = 890) ended in conviction of the offender and 33.7% (n = 452) in acquittal of the accused.

The proportions of conviction of the offender were respectively 63.8% considering crimes of abuse and 69.3% considering crimes concerning sexual violence.

## Characteristics of victims and defendants

Victims were mostly women (89.5%) and Italian (67%). When only considering judgements concerning sexual violence, the victim was a woman in 96.8% of cases. Table 2 shows the distribution of victims according to selected characteristics. No statistically significant association emerged between age of the victim and a guilty verdict (in 920 cases, 68.6%, however, the age of the victim was not indicated in the sentence).

Regarding the victim's history of use of substances, positivity for alcohol in a judgment of conviction was found in 2.1% of cases and positivity for drugs in 1.7%, while in judgments of acquittal in 4.0% and 1.1% respectively. This difference was not statistically significant.

No association emerged between the legal outcome and nationality of the victim nor the victims' occupation. However, in particular, the subjects' occupation was omitted in most of the verdicts (see S1 Table).

The distribution of defendants is shown in Table 3. Considering the total series, a history of alcohol abuse was respectively 18.2% in the condemned and in 13.3% in acquittal; as well as drug abuse, they had positive histories respectively in 6.2% and 4.9%.

Considering the total series the OR of a guilty verdict were, in comparison with defendants reporting no addiction, 1.2 (0.9–1.7) for defendants reporting drug or alcohol addiction or

**Table 2. Distribution of victims according to age and addiction.**

| VICTIMS | Conviction | | | | | | Acquittal | | | | | | Unadjusted Odds ratio (95% CI) | | |
|---|---|---|---|---|---|---|---|---|---|---|---|---|---|---|---|
| | Total* | | SV | | DV | | Total* | | SV | | DV | | Total* | SV | DV |
| | n = 890 | | n = 259 | | n = 558 | | n = 452 | | n = 115 | | n = 317 | | | | |
| | n | % | n | % | N | % | n | % | n | % | N | % | | | |
| **Age** | | | | | | | | | | | | | | | |
| < = 18 yrs | 126 | 14.2 | 73 | 28.2 | 44 | 7.9 | 69 | 15.3 | 26 | 22.6 | 41 | 12.9 | 1+ | 1+ | 1+ |
| 19–35 yrs | 89 | 10 | 36 | 13.9 | 44 | 7.9 | 35 | 7.7 | 11 | 9.6 | 20 | 6.3 | 1.4 (0.9–2.3) | 1.2 (0.5–2.6) | 2.1 (1.0–4.0) |
| >36 yrs | 66 | 7.4 | 12 | 4.6 | 51 | 9.1 | 37 | 8.2 | 13 | 11.3 | 23 | 7.3 | 1.0 (0.6–1.6) | 0.3 (0.1–0.8) | 2.1 (1.0–4.0) |
| Information Missing | 609 | 68.4 | 138 | 53.3 | 419 | 75.1 | 311 | 68.8 | 65 | 56.5 | 233 | 73.5 | // | // | // |
| **Addiction** | | | | | | | | | | | | | | | |
| No | 673 | 75.6 | 189 | 73.0 | 417 | 74.7 | 328 | 72.6 | 75 | 65.2 | 238 | 75.1 | 1+ | 1+ | 1+ |
| Drug | 15 | 1.7 | 7 | 2.7 | 10 | 1.8 | 5 | 1.1 | 4 | 3.5 | 11 | 3.5 | 0.5 (0.3–1.0) | 1.3 (0.4–4.1) | 0.5 (0.2–1.2) |
| Alcohol | 19 | 2.1 | 6 | 2.3 | 8 | 1.4 | 18 | 4.0 | 0 | 0 | 3 | 1.0 | 1.5 (0.5–4.1) | | 1.5 (0.4–5.8) |
| Information Missing | 183 | 20.6 | 57 | 22 | 123 | 22.0 | 101 | 22.4 | 36 | 31.3 | 65 | 20.5 | // | // | // |

*Total sample include sexual violence crimes (SV), abuses against family members or cohabitants crimes (DV) and cases concerning both of them. + Reference category.

ludopathy, and 1.9 (1.1–3.3) more than one addiction. Considering domestic violence, the OR of the guilty verdict were in comparison with defendants reporting no addiction 1.7 (1.1–2.6) in those reporting drug or alcohol addiction and 3.2 (95%CI 1.7–6.0) more than one addiction.

**Table 3. Distribution of defendants according to addiction and criminal records.**

| DEFENDANTS | Conviction | | | | | | Acquittal | | | | | | Unadjusted Odds ratio (95% CI) | | | Adjusted OR (95% CI)[a] | | |
|---|---|---|---|---|---|---|---|---|---|---|---|---|---|---|---|---|---|---|
| | Total* | | SV | | DV | | Total* | | SV | | DV | | Total* | SV | DV | Total* | SV | DV |
| | n = 890 | | n = 259 | | n = 558 | | n = 452 | | n = 115 | | n = 317 | | | | | | | |
| | n | % | N | % | N | % | n | % | n | % | n | % | | | | | | |
| **Addiction** | | | | | | | | | | | | | | | | | | |
| No | 426 | 47.9 | 165 | 63.7 | 213 | 38.2 | 256 | 56.6 | 68 | 59.1 | 172 | 54.3 | 1+ | 1+ | 1+ | 1+ | 1+ | 1+ |
| Drug | 55 | 6.2 | 11 | 4.3 | 40 | 7.2 | 22 | 4.9 | 4 | 3.5 | 18 | 5.7 | 1.5 (1.1–2.2)£ | 1.2 (0.6–2.5)¥ | 2.0 (1.4–2.7) £ | 1.2 (0.9–1.7) £ | 1.4 (0.6–3.2) ¥ | 1.7 (1.1–2.6) £ |
| Alcohol | 162 | 18.2 | 17 | 6.6 | 135 | 24.2 | 60 | 13.3 | 8 | 7.0 | 50 | 15.8 | | | | | | |
| Ludopathy | 8 | 0.9 | 0 | - | 7 | 1.3 | 7 | 1.6 | 0 | - | 7 | 2.2 | | | | | | |
| More than one | 89 | 10 | 11 | 4.3 | 71 | 12.7 | 22 | 4.9 | 1 | 0.9 | 19 | 6 | 2.4 (1.5–4.0) | | 3.0 (1.8–5.2) | 1.9 (1.1–3.3) | | 3.2 (1.7–6.0) |
| Information Missing | 150 | 16.9 | 55 | 21.2 | 92 | 16.5 | 85 | 18.8 | 34 | 29.6 | 51 | 16.1 | // | // | // | // | // | // |
| **Criminal record** | | | | | | | | | | | | | | | | | | |
| No | 554 | 62.3 | 156 | 60.2 | 337 | 60.4 | 281 | 62.2 | 65 | 56.5 | 201 | 63.4 | 1+ | 1+ | 1+ | 1+ | 1+ | 1+ |
| Yes | 96 | 10.8 | 24 | 9.3 | 65 | 11.7 | 20 | 4.4 | 5 | 4.4 | 15 | 4.7 | 2.4 (1.5–4.0) § | 2.0 (0.7–5.5) § | 2.6 (1.4–4.7) § | 2.2 (1.2–3.8) § | 1.6(0.5–4.7) § | 2.1 (1.1–4.3) § |
| Another type | 12 | 1.4 | 1 | 0.4 | 12 | 2.2 | 1 | 0.2 | 0 | | 1 | 0.3 | | | | | | |
| Information Missing | 240 | 27 | 79 | 30.5 | 156 | 28 | 151 | 33.4 | 45 | 39.1 | 101 | 31.9 | // | // | // | // | // | // |

*Total sample include sexual violence crimes (SV), abuses against family members or cohabitants crimes (DV) and cases concerning both of them. + Reference category.

[a]OR adjusted as specified in Material and Methods.

£ Drug, alcohol, ludopathy versus No.

¥ Drug, alcohol, ludopathy, more than one versus No.

§ Yes, another type versus No.

The percentage of perpetrators is higher both for those previously convicted of similar crimes and for those convicted of other types of crimes. Considering the total series, the risk of a guilty verdict was 2.2 (1.2–3.8) for defendants reporting a previous guilty verdict for any crime. Considering sexual and domestic violence, corresponding values were respectively 1.6 (0.5–4.7) and 2.1 (1.1–4.3).

No association emerged between defendant nationality and legal outcome (S2 Table). Likewise, no significant differences in the relationship between victims and defendants were detected. The most represented classes are husbands and cohabitants (respectively 31.1% and 16.7% in conviction and 30.1% and 17.3% in acquittal verdicts), followed by other family members (23% and 27.7% respectively) and finally by friends (13.4% and 12.4%) (S3 Table).

## Information on the crime

The predominant source of the crime report was the victim complaint, without differences between the legal outcomes (81.7% vs 84.3%) (Table 4).

In cases where a precautionary measure was applied (such as the removal of the alleged offender from the family home, etc.), 33.7% of the cases resulted in a conviction, compared to 12.2% of acquittals (OR of guilty verdict 2.3 95%CI 1.6–3.3 for the total series).

The application of personal safety measures (for example the removal of the victim and his/her protection in a secure facility), resulted in a guilty verdict for sexual violence in 7.7% of conviction, compared to 1.7% of acquittals.

The types of evidence most valued in the judgments seemed to be the declaration of the victim, medical certifications and depositions; the combination of multiple types of evidence led to an increase in convictions compared to acquittals: this association was statistically significant in the total series and in case of domestic violence. The presence of a medical certificate/other deposition was also associated with an OR of guilty verdict in case of sexual violence (OR 5.9, 95%CI 1.2–28.6). In the total series, the OR of guilty verdict was, in comparison with a duration of 36 months of the criminal proceedings, higher in case of duration <12 months (OR 2.3, 95%CI 1.5–3.5) and 12–22 months (OR 2.0, 95%CI 1.4–2.9).

The request for presentation as a civil party in criminal proceedings was associated with an increase of guilty verdict OR 2.6 (95%CI 1.9–3.4) for total series, OR 2.5 (95%CI 1.7–3.6) for domestic violence and 2.0 (95%CI 1.2–3.6) for sexual violence.

## Characteristics of the assaults

The characteristics of the assaults are considered in Table 5.

In most of the episodes there was only one defendant (95.5%); number of the assailants resulted not related to the outcome of the sentence. The presence of a minor witnessing the violence was reported in 42.5% of the cases in convictions and 35.2% in acquittals, this difference resulted not statistically significant.

The use of firearms or other weapons was associated with a guilty verdict in the total series (OR 2.5, 95%CI 1.4–4.5) and in case of domestic violence (OR 4.4, 95%CI 2.0–9.3).

Finally, it was observed that the more frequently the violence occurred, the more likely a sentence of conviction was issued: considering the total series, the OR was 3.3 (95%CI: 1.7–6.5) for daily episodes and 2.3 (95% CI: 1.2–4.6) for weekly episodes.

## Medical information

In most cases in which the data was known, the victim did not ask for medical assistance (57.5%). In 13.9% of the cases the victim went to SVSeD, while in 28.6% of the cases the victim went to a non-specialized emergency room. In cases of final acquittal, 61.7% of the victims did

**Table 4. Distribution of characteristics of the crime.**

| | Conviction | | | | | | Acquittal | | | | | | Unadjusted Odd Ratio (95% CI) | | | Adjusted Odd Ratio (95% CI)[a] | | |
|---|---|---|---|---|---|---|---|---|---|---|---|---|---|---|---|---|---|---|
| | Total* n | Total* % | SV N | SV % | DV N | DV % | Total* n | Total* % | SV n | SV % | DV n | DV % | Total* | SV | DV | Total* | SV | DV |
| **Source of the crime report** | | | | | | | | | | | | | | | | | | |
| Other | 151 | 17.0 | 48 | 18.5 | 91 | 16.3 | 63 | 13.9 | 14 | 12.2 | 47 | 14.8 | 1+ | 1+ | 1+ | | | |
| Complaint | 727 | 81.7 | 207 | 79.9 | 460 | 82.4 | 381 | 84.3 | 101 | 87.8 | 263 | 83 | 0.8 (0.6–1.1) | 0.6 (0.3–1.2) | 0.9 (0.6–1.3) | | | |
| Missing | 12 | 1.3 | 4 | 1.5 | 7 | 1.3 | 8 | 1.8 | 0 | - | 7 | 2.2 | // | // | // | | | |
| **application of precautionary measures** | | | | | | | | | | | | | | | | | | |
| No | 544 | 61.1 | 147 | 56.8 | 352 | 63.1 | 356 | 78.8 | 93 | 80.9 | 246 | 77.6 | 1+ | 1+ | 1+ | 1+ | 1+ | 1+ |
| yes | 300 | 33.7 | 94 | 36.3 | 178 | 31.9 | 55 | 12.2 | 12 | 10.4 | 40 | 12.6 | 3.6 (2.6–4.9) | 5.0 (2.6–9.5) | 3.1 (2.1–4.5) | 2.3 (1.6–3.3) | 4.4 (2.0–9.4) | 1.8 (1.1–2.8) |
| Missing | 46 | 5.2 | 18 | 7.0 | 28 | 5.0 | 41 | 9.1 | 10 | 8.7 | 31 | 9.8 | // | // | // | // | // | // |
| **application of personal security measures** | | | | | | | | | | | | | | | | | | |
| No | 768 | 86.3 | 213 | 82.2 | 490 | 87.8 | 389 | 86.1 | 102 | 88.7 | 268 | 84.5 | 1+ | 1+ | 1+ | | | |
| Yes | 56 | 6.3 | 20 | 7.7 | 28 | 5.0 | 19 | 4.2 | 2 | 1.7 | 16 | 5 | 1.5 (0.9–2.5) | 4.8 (1.1–20.9) | 1.0 (0.5–1.8) | | | |
| Missing | 66 | 7.4 | 26 | 10.0 | 40 | 7.2 | 44 | 9.7 | 11 | 9.6 | 33 | 10.4 | // | // | // | | | |
| **type of evidence** | | | | | | | | | | | | | | | | | | |
| Deposition of the victim | 222 | 24.9 | 80 | 30.9 | 132 | 23.7 | 215 | 47.6 | 69 | 60.0 | 139 | 43.9 | 1+ | 1+ | 1+ | 1+ | 1+ | 1+ |
| Medical certificate | 2 | 0.2 | 0 | - | 2 | 0.4 | 1 | 0.1 | 0 | - | 1 | 0.3 | | | | | | |
| Other deposition | 34 | 3.8 | 13 | 5.0 | 19 | 3.4 | 25 | 5.5 | 3 | 2.6 | 20 | 6.3 | 1.3 (0.8–2.3) | 3.7 (1.0–13.7) | 1.1 (0.6–2.0) | 1.2 (0.6–2.1) | 5.9 (1.2–28.6) | 1.0 (0.5–2.2) |
| Deposition of the victim + another of the previous ones | 407 | 45.7 | 136 | 52.5 | 242 | 43.4 | 148 | 32.7 | 36 | 31.3 | 104 | 32.8 | 2.7 (2.0–3.5) | 3.3 (2.0–5.3) | 2.5 (1.8–3.4) | 2.1 (1.6–2.9) | 2.0 (1.1–3.6) | 2.1 (1.4–3.1) |
| All of the previous | 218 | 24.5 | 27 | 10.4 | 159 | 28.5 | 56 | 12.4 | 5 | 4.4 | 49 | 15.5 | 3.8 (2.7–5.3) | 4.7 (1.7–12.8) | 3.4 (2.3–5.1) | 2.8 (1.8–4.3) | 4.0 (1.4–11.8) | 2.7 (1.7–4.4) |
| Missing | 7 | 0.8 | 3 | 1.2 | 4 | 0.7 | 7 | 1.6 | 2 | 1.7 | 4 | 1.3 | // | // | // | // | // | // |
| **Duration of the proceeding** | | | | | | | | | | | | | | | | | | |
| <12 months | 242 | 27.2 | 55 | 21.2 | 168 | 30.1 | 66 | 14.6 | 17 | 14.8 | 48 | 15.1 | 1+ | 1+ | 1+ | 1+ | 1+ | 1+ |
| 12–22 months | 248 | 27.9 | 87 | 33.6 | 144 | 25.8 | 84 | 18.6 | 17 | 14.8 | 66 | 20.8 | 2.6 (1.8–3.7) | 1.9 (1.0–3.8) | 2.8 (1.8–4.3) | 2.3 (1.5–3.5) | 1.4 (0.6–3.2) | 3.0 (1.8–5.0) |
| 23–35 months | 174 | 19.6 | 49 | 18.9 | 110 | 19.7 | 148 | 32.7 | 38 | 33.0 | 105 | 33.1 | 2.1 (1.5–2.9) | 3.1 (1.6–6.0) | 1.8 (1.2–2.6) | 2.0 (1.4–2.9) | 2.9 (1.4–5.9) | 1.6 (1.0–2.6) |
| >36 months | 189 | 21.2 | 60 | 23.2 | 107 | 19.2 | 134 | 29.7 | 36 | 31.3 | 86 | 27.1 | 0.8 (0.6–1.1) | 0.8 (0.4–1.4) | 0.8 (0.6–1.2) | 0.9 (0.6–1.3) | 0.8 (0.4–1.5) | 0.8 (0.5–1.3) |
| Missing | 37 | 4.2 | 8 | 3.1 | 29 | 5.2 | 20 | 4.4 | 7 | 6.1 | 12 | 3.8 | // | // | // | // | // | // |
| **Civil party** | | | | | | | | | | | | | | | | | | |
| No | 515 | 57.9 | 146 | 56.4 | 344 | 61.7 | 349 | 77.2 | 83 | 72.2 | 255 | 80.4 | 1+ | 1+ | 1+ | 1+ | 1+ | 1+ |
| Yes | 366 | 41.1 | 109 | 42.1 | 209 | 37.5 | 98 | 21.7 | 31 | 27.0 | 58 | 18.3 | 2.5 (1.9–3.3) | 2.0 (1.2–3.2) | 2.7 (1.9–3.7) | 2.6 (1.9–3.4) | 2.0 (1.2–3.6) | 2.5 (1.7–3.6) |
| Missing | 9 | 1.0 | 4 | 1.5 | 5 | 0.9 | 5 | 1.1 | 1 | 0.9 | 4 | 1.3 | // | // | // | // | // | // |

*Total sample include sexual violence crimes (SV), abuses against family members or cohabitants crimes (DV) and cases concerning both of them. + Reference category.

[a]OR adjusted as specified in Material and Methods.

**Table 5. Distribution of characteristics of the assault.**

| | Conviction | | | | | | Acquittal | | | | | | Unadjusted OR (95% CI) | | | Adjusted OR (95% CI)[a] | | |
|---|---|---|---|---|---|---|---|---|---|---|---|---|---|---|---|---|---|---|
| | Total* | | SV | | DV | | Total* | | SV | | DV | | Total* | SV | DV | Total* | SV | DV |
| | n | % | n | % | N | % | n | % | n | % | n | % | | | | | | |
| **Number of suspects** | | | | | | | | | | | | | | | | | | |
| 1 | 857 | 96.4 | 249 | 96.5 | 543 | 97.3 | 425 | 94.4 | 108 | 94.7 | 298 | 94.3 | 1+ | 1+ | 1+ | | | |
| 2 or more | 32 | 3.6 | 9 | 3.5 | 15 | 2.7 | 25 | 5.6 | 6 | 5.3 | 18 | 5.7 | 0.6 (0.-1.1) | 0.7 (0.2–1.9) | 0.5 (0.2–0.9) | | | |
| **Presence of minors** | | | | | | | | | | | | | | | | | | |
| Yes | 378 | 42.5 | 35 | 13.5 | 299 | 53.6 | 159 | 35.2 | 16 | 13.9 | 135 | 42.6 | 1.4 (1.1–1.7) | 1.0 (0.5–1.9) | 1.6 (1.2–2.1) | | | |
| No | 480 | 53.9 | 211 | 81.5 | 241 | 43.2 | 278 | 61.5 | 95 | 82.6 | 171 | 53.9 | 1+ | 1+ | 1+ | | | |
| Information Missing | 32 | 3.6 | 13 | 5 | 18 | 3.2 | 15 | 3.3 | 4 | 3.5 | 11 | 3.5 | // | // | // | | | |
| **Place of the offense** | | | | | | | | | | | | | | | | | | |
| own house | 610 | 68.6 | 81 | 31.3 | 466 | 83.5 | 335 | 74.1 | 41 | 35.7 | 281 | 88.6 | 0.7 (0.6–0.9) | 0.8 (0.5–1.3) | 0.5 (0.3–0.9) | | | |
| Other places | 262 | 29.4 | 176 | 68 | 77 | 13.8 | 103 | 22.8 | 72 | 62.6 | 25 | 7.9 | 1+ | 1+ | 1+ | | | |
| Information Missing | 18 | 2.0 | 2 | 0.8 | 15 | 2.7 | 14 | 3.1 | 2 | 1.7 | 11 | 3.5 | // | // | // | | | |
| **Use of weapons** | | | | | | | | | | | | | | | | | | |
| No | 631 | 70.9 | 184 | 71 | 381 | 68.3 | 329 | 72.8 | 73 | 63.5 | 239 | 75.4 | 1+ | 1+ | 1+ | 1+ | 1+ | 1+ |
| Yes | 92 | 10.3 | 20 | 7.7 | 67 | 12 | 17 | 3.8 | 5 | 4.4 | 10 | 3.2 | 2.8 (1.7–4.8) | 1.6 (0.6–4.3) | 4.2 (2.1–8.3) | 2.5 (1.4–4.5) | 1.0(0.3–3.2) | 4.4 (2.0–9.3) |
| Information Missing | 167 | 18.8 | 55 | 21.2 | 110 | 19.7 | 106 | 23.5 | 37 | 32.2 | 68 | 21.5 | // | // | // | // | // | // |
| **Frequency of episodes** | | | | | | | | | | | | | | | | | | |
| Daily | 207 | 23.2 | 9 | 3.5 | 174 | 31.2 | 35 | 7.7 | 2 | 1.7 | 32 | 10.1 | 3.6 (2.0–6.5) | 10.5 (1.6–70.8) § | 2.8 (1.4–5.4) | 3.3 (1.7–6.5) | 13.7 (1.5–121.6) § | 2.7 (1.3–5.6) |
| Weekly | 131 | 14.7 | 12 | 4.6 | 100 | 17.9 | 31 | 6.9 | 3 | 2.6 | 20 | 6.3 | 2.6 (1.4–4.8) | | 2.6 (1.2–5.3) | 2.3 (1.2–4.6) | | 2.3 (1.1–5.2) |
| Monthly | 43 | 4.8 | 2 | 0.8 | 39 | 7.0 | 26 | 5.8 | 5 | 4.4 | 20 | 6.3 | 1+ | 1+ | 1+ | 1+ | 1+ | 1+ |
| Another | 375 | 42.1 | 176 | 68.0 | 171 | 30.7 | 253 | 55.9 | 66 | 57.4 | 178 | 56.2 | 0.9 (0.5–1.5) | 6.7(1.3–35.2) | 0.5 (0.3–0.9) | 1.1 (0.6–2.0) | 13.2(1.9–88.9) | 0.5 (0.3–1.0) |
| Information Missing | 134 | 15.1 | 60 | 23.2 | 74 | 13.3 | 107 | 23.7 | 39 | 33.9 | 67 | 21.1 | // | // | // | // | // | // |

*Total sample include sexual violence crimes (SV), abuses against family members or cohabitants crimes (DV) and cases concerning both of them. + Reference category.

[a]OR adjusted as specified in Material and Methods.

§ Daily, weekly versus monthly.

not receive any medical assistance, in 11.1% of the cases they went to the SVSeD center, while in 27.2% of the cases to a general emergency room. No relation emerged between the type of medical assistance and a guilty verdict. Likewise no association emerged between a guilty verdict and penetration, prognosis and need of surgical intervention.

In 44.3% of cases no injuries were reported. Otherwise, multiple types of injuries were found in 15.6% of convictions and in 8.6% of acquittal cases.

The most affected site resulted the head, followed by the upper limbs, but in most of the cases injuries were observed in multiple sites. Considering the victims with one lesion, 296 of the total

series, the lesions most represented were bruises (79.7% n = 188), followed by lacerations (7.6% n = 18), abrasions (5.9% n = 14), fractures (5.5% n = 13) and burns (1.3% n = 3) (Table 6).

In cases where penetration was ascertained, SVSeD analysis observed injuries in 60.2% cases, against 43.7% of cases in which medical examination was performed in a non-specialized setting.

## Discussion

The current retrospective study analyzed cases that reached the final judgment in the Court of First Instance for sexual and domestic violence. In 33.7% there was the acquittal of the criminal

**Table 6. Distribution of medical information.**

| | Conviction | | | | | | Acquittal | | | | | | Unadjusted OR (95% CI) | | |
|---|---|---|---|---|---|---|---|---|---|---|---|---|---|---|---|
| | Total* | | SV | | DV | | Total* | | SV | | DV | | Total* | SV | DV |
| | N | % | n | % | n | % | n | % | n | % | n | % | | | |
| **SVSeD** | | | | | | | | | | | | | | | |
| No | 493 | 55.4 | 187 | 72.2 | 277 | 49.6 | 279 | 61.7 | 87 | 75.7 | 182 | 57.4 | 1+ | 1+ | 1+ |
| Yes | 136 | 15.2 | 46 | 17.8 | 69 | 12.4 | 50 | 11.1 | 17 | 14.8 | 28 | 8.8 | 1.5 (1.1–2.2) | 1.3 (0.7–2.3) | 1.6 (1.0–2.6) |
| Another emergency room | 261 | 29.3 | 26 | 10 | 212 | 38.0 | 123 | 27.2 | 11 | 9.6 | 107 | 33.8 | 1.2 (0.9–1.6) | 1.1 (0.5–2.3) | 1.3 (1.0–1.8) |
| **Penetration** | | | | | | | | | | | | | | | |
| No | 683 | 76.6 | 171 | 66.0 | 494 | 88.5 | 364 | 80.5 | 74 | 64.4 | 282 | 89.0 | 1+ | 1+ | 1+ |
| Yes | 125 | 14 | 66 | 25.5 | 7 | 1.3 | 46 | 10.2 | 31 | 27.0 | 4 | 1.3 | 1.5 (1.0–2.1) | 0.9 (0.6–1.5) | 1.0 (0.3–3.4) |
| Information Missing | 83 | 9.3 | 22 | 8.5 | 57 | 10.2 | 42 | 9.3 | 10 | 8.7 | 31 | 9.8 | // | // | // |
| **Prognosis** | | | | | | | | | | | | | | | |
| <6 | 153 | 17.2 | 51 | 19.7 | 92 | 16.5 | 97 | 21.5 | 22 | 19.1 | 69 | 21.8 | 1+ | 1+ | 1+ |
| > = 6 | 174 | 19.6 | 16 | 6.2 | 134 | 24.0 | 75 | 16.6 | 3 | 2.6 | 68 | 21.5 | 1.5 (1.0–2.1) | 2.3 (0.6–8.7) | 1.5 (1.0–2.3) |
| Information Missing | 564 | 63.3 | 192 | 74.1 | 332 | 59.5 | 280 | 62 | 90 | 78.3 | 180 | 56.8 | // | // | // |
| **Surgical interventions** | | | | | | | | | | | | | | | |
| No | 689 | 77.4 | 195 | 75.3 | 426 | 76.3 | 340 | 75.2 | 80 | 69.6 | 243 | 76.7 | 1+ | 1+ | 1+ |
| Yes | 20 | 2.3 | 6 | 2.3 | 11 | 2.0 | 8 | 1.8 | 0 | - | 5 | 1.6 | 1.2 (0.5–2.8) | - | 1.3 (0.4–3.7) |
| Information Missing | 181 | 20.3 | 58 | 22.4 | 121 | 21.7 | 104 | 23.0 | 35 | 30.4 | 69 | 21.8 | // | // | // |
| **Injuries** | | | | | | | | | | | | | | | |
| No | 370 | 41.6 | 148 | 57.1 | 194 | 35.5 | 224 | 49.6 | 71 | 61.7 | 144 | 45.4 | 1+ | 1+ | 1+ |
| One type | 199 | 22.4 | 26 | 10.4 | 149 | 26.7 | 97 | 21.5 | 8 | 6.7 | 82 | 25.9 | 1.2 (0.9–1.7) | 2.4 (1.1–5.2) | 1.4 (1.0–1.9) |
| More than one | 139 | 15.6 | 19 | 7.3 | 101 | 18.1 | 39 | 8.6 | 1 | 0.9 | 34 | 10.7 | 2.2 (1.5–3.2) | | 2.2 (1.4–3.4) |
| Information Missing | 182 | 20.5 | 66 | 25.5 | 111 | 20.0 | 92 | 20.4 | 35 | 30.4 | 57 | 18.0 | // | // | // |
| **Injury (location)** | | | | | | | | | | | | | | | |
| No | 371 | 41.7 | 149 | 57.5 | 194 | 34.8 | 224 | 49.6 | 71 | 61.7 | 74 | 23.3 | 1+ | 1+ | 1+ |
| Head | 115 | 12.9 | 19 | 7.3 | 82 | 14.7 | 56 | 12.4 | 5 | 4.4 | 48 | 15.1 | 1.2 (0.9–1.8) | 2.0(0.9–4.3)# | 1.3 (0.8–1.9) |
| Neck | 11 | 1.2 | 0 | - | 7 | 1.3 | 7 | 1.6 | 0 | - | 7 | 2.2 | 1.0 (0.7–1.5) § | | 1.1 (0.7–1.8) |
| Chest | 12 | 1.4 | 1 | 0.4 | 10 | 1.8 | 8 | 1.8 | 1 | 0.9 | 5 | 1.6 | | | |
| Abdomen | 8 | 0.9 | 2 | 0.8 | 5 | 0.9 | 3 | 0.7 | 1 | 0.9 | 2 | 0.6 | | | |
| Upper limbs | 30 | 3.4 | 3 | 1.2 | 25 | 4.5 | 18 | 4.0 | 0 | - | 18 | 5.7 | | | |
| Lower limbs | 8 | 0.9 | 2 | 0.8 | 5 | 0.9 | 5 | 1.1 | 1 | 0.9 | 3 | 0.9 | | | |
| Multiple sites | 145 | 16.3 | 10 | 3.9 | 116 | 20.8 | 40 | 8.9 | 1 | 0.9 | 36 | 11.4 | 2.2 (1.5–3.2) | | 2.4 (1.6–3.7) |
| Information Missing | 190 | 21.4 | 73 | 28.2 | 114 | 13 | 91 | 20.1 | 35 | 30.4 | 54 | 17.0 | // | // | // |

*Total sample include sexual violence crimes (SV), abuses against family members or cohabitants crimes (DV) and cases concerning both of them. + Reference category

§ Neck, chest, abdomen, upper limbs, lower limbs versus No.

# Neck, chest, abdomen, upper limbs, lower limbs, multiple sites versus No.

proceeding, while for 66.3% there was a conviction. Convictions are higher than in previously published works [16]. The large number of convictions in our population must be related to the fact that we analyzed cases which actually went to court, and therefore did not deal with cases which had been dismissed by the prosecutor at the end of the preliminary investigations.

Previous studies considering this type of data [16, 21–23] had no diriment results, since none of the variable studied (results of forensic samples, ano-genital lesions, extragential lesions. . .) proved to be predictive [17, 18]. Also Saint-Martin et al. [16] failed to demonstrate any association between medical findings and convictions. However documentation of multiple injuries has been previously associated to judgments indicating non-consensual compared to consensual sexual activity [24, 25]. Jewkes et al observed that documentation of anogenital injuries was not associated with charge filing and arrest of the alleged assailant but, on the other hand, increased the likelihood for conviction in court [19].

Our study observed that medical examination is relevant, although it must be considered in a wider context of evidence. Indeed, a factor influencing the judge's decision of conviction or acquittal was identified in the presence of different sources of evidences. Assistance in a specialized medical center was more represented in the group of convictions, noting the usefulness of the formation of objective medical elements of judgment in aid of the magistrate's decision. However, the main source of information in the present study was represented by the declarations of the victims; only in 0.2% of cases medical information was the only source of evidence on the basis of which the verdict was pronounced. Regardless it has to be stressed that medical findings if properly interpreted can give clues on the consistency between the history told and injuries observed, and thus reveal a number of elements pertaining to credibility; even if "it's normal to be normal" [26] and previous papers documented that physical injuries have not been associated with charges [13]. Indeed, a pivotal part of the physician's work also lies in explaining this phenomenon: that also no lesions can be present in such kinds of abuses. However, in most of the cases where the data was known, the victim did not ask for medical assistance (57.5%), raising reflections on the need to implement the accessibility of these services and the importance of promoting them. About that, in cases examined in the SVSeD, 52.4% were observed in the first 72 hours of the assault (of these, the average was 10.2 hours), while in cases not examined in the SVSeD, but just in a general emergency ward, only 17.6% sought attention in the first 72 hours (average 5.5 hours). In cases where penetration was ascertained, SVSeD analysis observed injuries in 60.2% cases, against 43.7% of cases in which medical examination was performed in a non-specialized setting, confirming the relevance of medical examination expertise in identifying and characterizing the presence of injuries. Data concerning performed medico-legal sample analysis, such as DNA and toxicological tests, were not reported in any of the sentences, suggesting the non-execution or the irrelevance of the data in the sentence motivation.

In most of the cases, the assailant was an acquaintance, a compatriot, and had no previous criminal records. As observed by Saint-Martin et al. [16] today most of the cases are related to acquaintances than to strangers. However, nationality and victim's acquaintance of the assailant does not appear to have significantly affected the judge's decision in our population.

If the victim was under the age of 35, the likelihood for conviction of the alleged perpetrator increased, probably due to a greater tendency to report the fact by the victim: such an element may be stressed in Public Health programs in order to encourage victims to report.

History of alcohol and/or drug use of the victim presented no statistically significant difference in conviction and acquittal cases. On the contrary, the defendant's history of use of alcohol or illicit drugs considerably increased the likelihood for the judge issuing a sentence against the offender, especially if combined. Therefore, this habit may be regarded as an aggravating factor in the crime but it also may be the expression of a wider disadvantage of the defendant.

The likelihood for conviction was also increased if the offender was recidivist (already known to the Court), whether the defendant had received a previous conviction for the same type of crime (sexual violence, etc.), but also if the previous crime was different (other crimes against the person, drug trafficking, etc.). In cases where a precautionary measure was applied to the victim (e.g transfer of the victim in a protected area, restraining order, etc.) the Court expressed a judgment of conviction in most of the cases. In these circumstances the crime was often characterized by increased violence and recidivism. The same reasoning may be applied in cases where a weapon was used during the assault or in cases where violence was repeated on several occasions on the same victim (the higher the frequency, the greater the frequency of conviction is).

Cases with observed multiple types of lesions, especially if registered in multiple anatomical areas, were characterized by conviction, probably because they suggest a greater violence of the assailant, as previously seen when examining motivations sustaining the assignment of humanitarian protection to asylum seekers [27]. Actually, a higher number of convictions were registered in cases where the victim had clinical documentation produced by a general emergency room or a specific anti-violence service (as in the case of SVSeD), suggesting the importance that objective medical examinations may have. Concerning this aspect, however, it has to be stressed that in the whole studied population the correlation between number of days of prognosis and conviction could not be performed, given the limited data obtained on this aspect (due to high number of missing value).

Victim's history of alcohol intake appears to negatively affect the judge's ruling, increasing acquittal cases. This could be related to the increase of cases with no memory of the events by the victim and implicating the absence of a valid victim's description of the event, as well as scarcity of possible evidence that could be obtained during the investigation. Indeed, half of these acquittal cases were based only on the declaration of the victim, highlighting the need for further evidence in order to allow the court to express a guilty verdict.

The study shows that the reporting mode of the crime does not affect the likelihood for a conviction or acquittal, therefore it does not appear relevant if the source of the report is the police, the victim or other. Regarding the duration of the proceedings, lengths of the trial >22 months appeared more prone to final acquittal, which may be explained by the difficulty of having and presenting valid evidence for judicial purposes.

## Limits of the study

The data of this study was obtained from the sentences of the Court of First Instance. The lack of thorough data is one of the limits of this retrospective study, given that not all of them were explicitly stated in the verdicts, due to privacy issues. Moreover, in this work all the cases that have not been judged by the Court of First Instance have not been considered, therefore the results are not a real mirror of all the cases observed in the territory: all those crimes for which the offender asked for a shortened procedure are also absent as well as those that never reached trial. The same for summary judgement. In addition, it must be considered that the population examined is a sample of the Milan metropolitan area. Therefore, the data of this study are representative of a large metropolitan centre, which may be different from a smaller and more rural reality.

## Conclusions

Over a five-year period (from 2011 to 2015) 1342 judgments were issued by the Court of First Instance in Milan for such offences. Typically, the defendants were compatriots and acquaintances. Cases of conviction were more frequent when they involved the presence of clinical

documentation together with other deposition in addition to victim's deposition; the duration of proceeding less 22 months and the request for presentation as a civil party; the use of a weapon by the assailant, as well as if the assailant had a criminal record and had a history of drug abuse or other addictions. Frequent episodes of violence and application of precautionary measures were also associated to conviction. Therefore, many factors seem able to influence the judge's judgment, including medical data on number, type and injury characteristics, although clearly each case must be singularly evaluated. The mere presence of medical documentation, without the support of other sources of evidence, such as the victim's statement or further declarations, however, is almost always not definitive for the verdict. Despite so, in cases where there are multiple sources of evidence, clinical documentation can provide useful elements.

## Supporting information

**S1 Table. Distribution of victims according to nationality and occupation.**
(DOCX)

**S2 Table. Distribution of defendants according to nationality.**
(DOCX)

**S3 Table. Distribution of relationship between victims and defendants.**
(DOCX)

## Acknowledgments

Authors are very grateful to the Court of Milan for making the sentences available anonymously.

## Author Contributions

**Conceptualization:** Alberto Blandino, Federica Collini, Patrizio Nicolò, Giussy Barbara, Alessandra Kustermann, Cristina Cattaneo, Andrea Gentilomo.

**Data curation:** Alberto Blandino, Lidia Maggioni, Fabio Parazzini, Daniele Capano, Elena Maria Florio, Manuela Margherita, Gian Marco Bertelle, Lorenzo Franceschetti, Alberto Amadasi, Giulia Vignali, Barbara Ciprandi, Graziano Domenico Luigi Crudele, Vera Gloria Merelli, Enrico Angelo Muccino.

**Formal analysis:** Francesca Chiaffarino, Fabio Parazzini, Andrea Gentilomo.

**Methodology:** Alberto Blandino, Francesca Chiaffarino, Fabio Parazzini.

**Project administration:** Alberto Blandino, Cristina Cattaneo, Andrea Gentilomo.

**Software:** Francesca Chiaffarino, Fabio Parazzini.

**Supervision:** Alberto Blandino, Patrizio Nicolò, Cristina Cattaneo, Andrea Gentilomo.

**Validation:** Francesca Chiaffarino, Fabio Parazzini, Cristina Cattaneo, Andrea Gentilomo.

**Visualization:** Francesca Chiaffarino, Fabio Parazzini, Federica Collini, Alessandra Kustermann, Cristina Cattaneo, Andrea Gentilomo.

**Writing – original draft:** Alberto Blandino, Lidia Maggioni.

**Writing – review & editing:** Alberto Blandino, Lidia Maggioni, Fabio Parazzini, Federica Collini, Alessandra Kustermann, Cristina Cattaneo, Andrea Gentilomo.

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
