## [Decision Letter · Decision Letter 0]

14 Apr 2021

PONE-D-20-40869

Sexual assault and  abuse committed against family members: an analysis of 1342 legal outcomes and their motivations

PLOS ONE

Dear Dr. Blandino,

Thank you for submitting your manuscript to PLOS ONE. After careful consideration, we feel that it has merit but does not fully meet PLOS ONE’s publication criteria as it currently stands. Therefore, we invite you to submit a revised version of the manuscript that addresses the points raised during the review process.

We look forward to receiving your revised manuscript.

Kind regards,

Neil J. Vincent, Ph.D.

Academic Editor

PLOS ONE

Journal Requirements:

Reviewers' comments:

Reviewer's Responses to Questions

**Comments to the Author**

1. Is the manuscript technically sound, and do the data support the conclusions?

Reviewer #1: Yes

2. Has the statistical analysis been performed appropriately and rigorously? 

Reviewer #1: No

3. Have the authors made all data underlying the findings in their manuscript fully available?

Reviewer #1: No

4. Is the manuscript presented in an intelligible fashion and written in standard English?

Reviewer #1: Yes

5. Review Comments to the Author

Reviewer #1: The authors of this paper use data on sexual assault and abuse cases involving family members to identify the predictors of case outcomes (i.e., conviction versus acquittal). The data include cases in which the Public Prosecutors Office in Milan (Italy) issued a judgment from 2011 to 2015. The data on each case are comprehensive and include factors (i.e., medical information) not usually available in a study of this type. Their focus in on the effects of medical information on case outcomes; they conclude that these medical indicators are not, in and of themselves, definitive predictors of whether the case would result in conviction or acquittal.

To assess the effects of victim characteristics, defendant characteristics, crime characteristics, and medical information on case outcomes, the authors conduct a series of logistic regression analyses addressing each set of independent variables. Typically, in a study like this, the analysis would include all of the independent variables in the model (either entering the variables in blocks or entering them all initially). Given that the authors' approach differs from this, they should provide some justification for not including all of the factors in the final model.

Overall, this is an interesting paper on outcomes in sexual assault and abuse cases in a context (Italy) that has not been subject to prior analysis.

6. PLOS authors have the option to publish the peer review history of their article (what does this mean?). If published, this will include your full peer review and any attached files.

Reviewer #1: **Yes: **Cassia Spohn

---

## [Author Response · Author response to Decision Letter 0]

24 May 2021

As required from the referee, we applied the backward stepwise regression method and reported the adjusted odds ratios and the relative 95% confidence interval of the final multivariate logistic regression model concerning the overall sample. To allow the comparison of the results, we include in the analyses of the two subgroups (sexual violence and domestic violence) the same variables of the final logistic regression model of the overall sample. Results have been modified accordingly.

---

## [Decision Letter · Decision Letter 1]

17 Jun 2021

Sexual assault and  abuse committed against family members: an analysis of 1342 legal outcomes and their motivations

PONE-D-20-40869R1

Dear Dr. Blandino,

We’re pleased to inform you that your manuscript has been judged scientifically suitable for publication and will be formally accepted for publication once it meets all outstanding technical requirements.

Kind regards,

Neil J. Vincent, Ph.D.

Academic Editor

PLOS ONE

Additional Editor Comments (optional):

Reviewers' comments:

Reviewer's Responses to Questions

**Comments to the Author**

1. If the authors have adequately addressed your comments raised in a previous round of review and you feel that this manuscript is now acceptable for publication, you may indicate that here to bypass the “Comments to the Author” section, enter your conflict of interest statement in the “Confidential to Editor” section, and submit your "Accept" recommendation.

Reviewer #1: All comments have been addressed

2. Is the manuscript technically sound, and do the data support the conclusions?

Reviewer #1: Yes

3. Has the statistical analysis been performed appropriately and rigorously? 

Reviewer #1: Yes

4. Have the authors made all data underlying the findings in their manuscript fully available?

Reviewer #1: Yes

5. Is the manuscript presented in an intelligible fashion and written in standard English?

Reviewer #1: Yes

6. Review Comments to the Author

Reviewer #1: (No Response)

7. PLOS authors have the option to publish the peer review history of their article (what does this mean?). If published, this will include your full peer review and any attached files.

Reviewer #1: No

---

## [Editor Report · Acceptance letter]

21 Jun 2021

PONE-D-20-40869R1 

Sexual assault and abuse committed against family members: an analysis of 1342 legal outcomes and their motivations 

Dear Dr. Blandino:

I'm pleased to inform you that your manuscript has been deemed suitable for publication in PLOS ONE. Congratulations! Your manuscript is now with our production department. 

Kind regards, 

on behalf of

Dr. Neil J. Vincent 

%CORR_ED_EDITOR_ROLE%

PLOS ONE